# Chemical Structure and Microscopic Morphology Changes of Dyed Wood Holocellulose Exposed to UV Irradiation

**DOI:** 10.3390/polym15051125

**Published:** 2023-02-23

**Authors:** Hui Shi, Yongqing Ni, Hongwu Guo, Yi Liu

**Affiliations:** 1National Forestry and Grassland Engineering Technology Center for Wood Resources Recycling, Beijing Forestry University, Beijing 100083, China; 2Jirong Furniture Co., Ltd., Cangzhou 062150, China; 3Beijing Key Laboratory of Wood Science and Engineering, Beijing Forestry University, Beijing 100083, China

**Keywords:** dyed wood holocellulose, UV irradiation, photoaging, chemical structure, microscopic morphology

## Abstract

Dyed wood is prone to photoaging when exposed to UV irradiation which decreases its decorative effect and service life. Holocellulose, as the main component of dyed wood, has a photodegradation behavior which is still unclear. To investigate the effect of UV irradiation on chemical structure and microscopic morphology changes of dyed wood holocellulose, Maple birch (*Betulacostata Trautv*) dyed wood and holocellulose were exposed to UV accelerated aging treatment; the photoresponsivity includes crystallization, chemical structure, thermal stability, and microstructure were studied. Results showed that UV radiation has no significant effect on the lattice structure of dyed wood fibers. The wood crystal zone diffraction 2*θ* and layer spacing was basically unchanged. With the UV radiation time extension, the relative crystallinity of dyed wood and holocellulose showed a trend of increasing first and then decreasing, but the overall change was not significant. The relative crystallinity change range of the dyed wood was not more than 3%, and the dyed holocellulose was not more than 5%. UV radiation caused the molecular chain chemical bond in the non-crystalline region of dyed holocellulose to break, the fiber underwent photooxidation degradation, and the surface photoetching feature was prominent. Wood fiber morphology was damaged and destroyed, finally leading to the degradation and corrosion of the dyed wood. Studying the photodegradation of holocellulose is helpful to understand the photochromic mechanism of dyed wood, and, further, to improve its weather resistance.

## 1. Introduction

Wood is favored for its beautiful and varied patterns and comfortable colors, and has been widely used in housing construction, furniture manufacturing and interior decoration. With the increasing attention paid to the protection of forests and human settlements, the supply of wood for natural forests and precious species is very scarce, while the color and texture of fast-growing wood for ordinary plantations and secondary forests often fail to meet the application requirements. Color is an important index to evaluate wood surface properties and commodity value. Wood dyeing is a new technology to realize the efficient development and utilization of low-quality and fast-growing plantation wood. Normally, the wood dyeing process mainly includes computer color matching, veneer color matching, dyeing treatment, computer three-dimensional modeling design, wood texture reorganization, gluing, planning and other links [1,2,3]. Wood dyeing can improve the surface visual characteristics and decorative effects of plantation wood. It can be used to imitate the texture of precious materials and create artistic patterns and handicrafts. These products have now achieved industrial-scale production.

Wood is a naturally growing heterogeneous porous biomass polymer with a developed and complex tubular pore network structure. Wood fiber contains rich hydrophilic groups (such as carboxyl and hydroxyl). Wood dyeing is a process in which the dye solution wets, diffuses and adsorbs on the wood surface, while the dye molecules penetrate into the wood interior through the large capillary channel, and then diffuses, settles and adsorbs on the wood fiber surface through the capillary channel, thus making the wood colored. However, like natural wood, dyed wood is always easily affected by light, heat, moisture and other environmental factors to fade during use, which significantly decreases its decorative effect and shortens its service life [4,5]. The photochromism of dyed wood differs from the independent system of wood and dyes. It results from the comprehensive action of multiple factors such as light, wood, dye and environment. It includes both the degradation of material molecules absorbing radiation and the oxidation of free radicals, which is a complex photochemical reaction process. The factors affecting the photochromism of dyed wood mainly include the chemical structure of wood, the variety of dyes, the combination mode of dye molecules and wood, the wavelength of radiation light, and the conditions of treatment medium [6,7,8,9].

Wood surface color is an important physical and chemical feature of wood. It is not only closely related to people’s visual and psychological feelings, but also reflects the composition and chemical structure of wood internal components. The main components of wood are cellulose, hemicellulose and lignin. Holocellulose is the general term for cellulose and hemicellulose. Generally, the holocellulose content in coniferous wood is 65~73%, and that in hardwood is about 70~82%. Wood components contain a large number of polar functional groups. Cellulose contains alcohol hydroxyl groups, lignin has many methoxy, carboxyl and phenolic hydroxyl groups, and hemicellulose has acetyl groups. These rich groups make the wood surface charged. Due to the different structures of different kinds of dye molecules, the binding forms and mechanisms between them and wood chemical groups are also different, which leads to different light fastness of dyed wood. Generally, the combination of acid and direct dyes with wood is mainly through hydrogen bonds or intermolecular van der Waals force, and the combination mode is mainly physical adsorption; Reactive dyes can react with wood components, so they can combine with wood through chemical bonds. Improving the binding strength between dye molecules and wood groups is conducive to improving the light fastness of dyed wood. In addition, because dyed wood is usually used for interior decoration and furniture products, the main role of light radiation in the short term is to make it discolor and fade, which has little impact on its physical and mechanical properties [10,11,12,13].

Nowadays, acid dyes have widely used in the wood dyeing industry as their good permeability, chemical stability, low price, comprehensive chromatography, mature dyeing process, and good dyeing effect. Generally, acid dyes are mainly used to dye wood lignin, but their dyeing ability to holocellulose is poor. Therefore, in the study of the effects of wood chemical components on the physical and chemical properties of dyed wood, little attention has been paid to holocellulose [13,14,15,16]. However, holocellulose, as the main chemical component of wood, could undergo a photodegradation reaction in long-term UV radiation, mainly including photodegradation and photosensitization. The energy gathered by short-wavelength UV light can dissociate the chemical bonds of cellulose, break the macromolecular chain of cellulose or lead to a light-emitting oxidation reaction, which may destroy the fiber morphology of dyed wood and then affect its processing and finishing properties.

In this paper, the dyed maple birch wood and holocellulose were used as test materials, and they were treated by UV accelerated aging chamber. The photodegradation behavior of the dyed wood holocellulose during light radiation was discussed, including crystallization property, chemical structure, thermal stability, microstructure, etc. Studying the chemical structure and microscopic morphology degradation behavior of dyed wood holocellulose is helpful in understanding the photochromic mechanism of dyed wood. It can provide theoretical support and guidance for preventing the light degradation of dyed wood for interior decoration and improving its weather resistance. This research has important potential ecological and economic benefits for reducing natural deforestation and realizing the high value-added utilization of artificial wood.

## 2. Materials and Methods

### 2.1. Materials

Fast-growing maple birch (*Betulacostata Trautv*) sapwood peeling veneer was purchased from Mudanjiang City, Heilongjiang Province, China, with a specification of 300 mm × 300 mm × 0.7 mm, the water content was 8%, and the air dry density was 0.607 g/cm^3^. Acid lake blue V dyes (*C. I. Acid Blue 1*), the chemical formula is C_27_H_33_N_2_NaO_6_S_2_, was provided by Tianjin No.2 Dye Chemical Factory. Its appearance is dark blue powder, which is easily soluble in water, has no tendency of aggregation at room temperature, and has bright and pure color, good levelness and wide application. It is a representative dye in wood dyeing. Anhydrous Na_2_SO_4_ was purchased from Beijing Chemical Plant with analytical purity; benzene, 95% ethanol, glacial acetic acid, and acetone were purchased from Beijing Lanyi Chemical Products Co., Ltd, Bejiing, China. with analytical purity; sodium chlorite, Tianjin Telong Biochemical Co., Ltd.,Tianjin, China, industrial grade, purity 82%.

### 2.2. Holocellulose Separation

Crush the maple birch wood veneer, screen the wood powder of 40~80 mesh, and extract it for 6 h (6 cycles per hour) by Soxhlet extraction with a benzene alcohol solution (V_benzene_:V_ethanol_ = 2:1). Weigh 20 g of the extracted sample into a 1000 mL beaker, and add 600 mL of distilled water for dispersion. Put the beaker into the water bath, raise the temperature to 75 °C, then add 5 mL glacial acetic acid and 6 g sodium chlorite (calculated by 100% purity) into the beaker every 1 h, stir it every 10 min, and heat it for 4 h. After treatment, filter with a G_2_ glass filter, repeatedly wash with distilled water until the washing solution is neutral, wash with acetone three times, dry the sample in a 105 °C oven for 24 h, and seal it for standby.

### 2.3. Dyeing Treatment

The maple birch wood veneer and holocellulose were dipped and dyed with acid lake blue V dyes under normal pressure. The dye content in the dye solution was 0.15%, and 0.15% (*w*/*v*) anhydrous Na_2_SO_4_ was added as a buffer. A 10% (*w*/*w*) H_2_SO_4_ solution was slowly added to the dye solution and the pH value of the dye solution adjusted to 4.0. Put the test piece into the stainless steel dye tank with the dye bath ratio of 1:15 (V_veneer_:V_dye solution_). Put the dye tank into an electric thermostatic water bath, raise the temperature to 90~95 °C, and maintain the constant temperature for 4 h. Stir the dye solution clockwise every 20 min to ensure uniform dyeing. Take out the test piece, wash off the floating color on the surface, and put it in a cool place to air dry to the moisture content of 8%.

### 2.4. UV Radiation Treatment

Crush the prepared dyed wood and screen 40~60 mesh of dyed wood powder. Weigh 1.500 g of dyed wood powder, dyed holocellulose, and acid lake blue V dyes, respectively, then lay them in a glass dish (Ø 120 mm) with a paving thickness of about 1 mm. A UV accelerated aging chamber was used for light radiation treatment. The light source use UVB-313 lamp tubes, and the light wavelength was concentrated at 313 nm. The distance between the lamp axis and the surface of the test piece is 14 cm, and the irradiance is 35 W/m^2^. The samples were taken out in batches after being irradiated for 0, 3, 5, 10, 20, 40, 60 and 100 h respectively, then dried for 12 h in a 55 °C oven, and sealed for standby.

### 2.5. Crystallization and Thermal Performance Analysis

The dyed wood powder and holocellulose were pressed into thin slices, and the relevant parameters of their crystallization properties were measured and calculated by X-ray diffractometer (XRD). The diffraction pattern was determined by continuous scanning mode, with the scanning range was 5.0~40.0°, the step length was 0.20°, the scanning speed was 2.0 °·min^−1^, the tube voltage was 40 kV, the tube current was 30 mA, and the sample table rotation speed was 30 r·min^−1^.

Differential scanning calorimeter (DSC) was used to test the thermal degradation characteristics of holocellulose, dyed holocellulose, and dyed holocellulose samples after 100 h of UV radiation. The endothermic peak of the sample between 20 °C and 215 °C was mainly discussed. The heating rate was 20 °C/min, and the nitrogen flow rate was 50 mL/min.

### 2.6. Chemical Structure and Microscopic Morphology Analysis

The chemical structure changes of dyed holocellulose and dyes before and after UV radiation were characterized by attenuated total reflectance Fourier transform infrared spectroscopy (ATR-FTIR). The spectral test wavelength was 4000~400 cm^−1^, the step length was 8 cm^−1^, and the scanning was 64 times. Scanning electron microscope (SEM) was used to characterize the microscopic morphology of dyed wood.

## 3. Results

### 3.1. Effect of UV Radiation on the Crystalline Properties of Dyed Wood

The XRD patterns of dyed wood and the relevant parameters of wood fiber crystallinity during UV radiation are shown in Figure 1 and Table 1. The crystal peaks of the XRD curves at the diffraction angle (2*θ*) near 17°, 22°, and 35° represent the diffraction peaks of (101), (002), and (040) crystalline surfaces of wood cellulose, respectively.

The diffraction peak intensity of dyed wood (101) and (002) decreased during UV radiation treatment (Figure 1), indicating that the crystalline zone relative content of the dyed wood changed under the action of short-wavelength UV light. However, the diffraction angle (2*θ*) corresponding to the crystalline area of dyed wood cellulose was basically maintained at about 22.1°, and the layer spacing *d*_(002)_ of the (002) crystal surface in the cellulose crystal cell was also kept at about 4.00 nm (Table 1). This indicated that the crystal structure of the cellulose crystal region was not damaged during the UV radiation process, and the structure of the cellulose crystal region was relatively stable to light. It can be speculated that the decrease of the diffraction peak intensity of dyed wood, and the change of the relative content of the crystalline area of dyed wood was mainly due to the oxidation or degradation of the non-crystalline area of the wood fiber. The oxidation degradation of the non-crystalline region of the wood fiber made the diffraction peak of the dyed wood decrease.

With the UV radiation time extension, the relative crystallinity and average grain size (*d*_(002)_) of dyed wood cellulose generally increased first and then decreased, but the overall change was not significant, and the change of relative crystallinity was no more than 3%. In the early stage of light radiation, the relative crystallinity of dyed wood increased. One reason was the relative content of crystalline area increased due to the oxidation degradation of the non-crystalline area of the wood fiber. In addition, under the action of UV light and heat, part of the absorbed water combined with hydroxyl hydrogen bonds in the wood cellulose non-crystalline area was possibly desorbed (Figure 2), making part of the non-crystalline area turn into crystalline area [17]. They result in an increase in the relative content and grain size of the crystallization zone. However, with the extension of UV radiation time, the chemical bonds on the cellulose molecular chain in the non-crystalline region continued to break due to photooxidation and gradually extended to the crystalline region, and the relative crystallinity of the wood fiber decreased. The wood fiber microfiber tow also slowly deteriorated from surface to inside due to photodegradation.

The acid substances or small molecules produced by acid dyes interacting with oxygen may also promote the photodegradation of dyed wood fibers, thus reducing the overall relative crystallinity of dyed wood. The specific evidence is that the diffraction peak intensity of dyed wood was decreased in Figure 1.

### 3.2. Effect of UV Radiation on the Relative Crystallinity of Dyed Holocellulose

The effect of UV radiation on the relative crystallinity of dyed holocellulose is shown in Figure 3. Due to a large amount of hydroxyl groups in the side chain of cellulose macromolecules, there is a strong hydrogen bond connection between cellulose molecules in the crystalline region, which makes dyed holocellulose have good chemical stability. It can be seen from Figure 3, the positions of diffraction peaks of dyed holocellulose (101) and (002) have not changed during the UV radiation treatment. This indicates the crystal area lattice structure of the dye holocellulose has not been damaged under UV light, and the dyed holocellulose was relatively stable to UV radiation. However, the diffraction peak intensity of the dyed holocellulose decreased slightly, and its relative crystallinity decreased with the extension of UV radiation time (0~100 h), but the total amount was not more than 5%. This indicates that the dyed holocellulose was degraded to a certain extent during UV radiation. These test results were consistent with the conclusions in Section 3.1.

### 3.3. Effect of UV Radiation on the Thermal Stability of Dyed Holocellulose

Comparative analysis of the holocellulose, dyed holocellulose, and dyed holocellulose after 100 h of UV aging (Figure 4), the endothermic peak (temperature) of the three samples presented an increasing trend, which was 122.75 °C, 124.30 °C, and 131.24 °C, respectively, but their endothermic capacity (heat flow) shows a decreasing trend. The endothermic peaks of the three samples can be attributed to the dissociation of cellulose adsorbed water. Because the vaporization latent heat of adsorbed water, and the hydrogen bond energy between water molecules and the free hydroxyl groups of wood microfilaments were large, the dissociation heat absorption was strong, resulting in a large endothermic peak.

After dyeing treatment, the endothermic peak temperature of wood holocellulose slightly increased. This phenomenon was mainly due to the reorientation of some cellulose molecular chains in the wood fiber bundle during the dyeing hydrothermal treatment process [12,13,14]. This induced the fiber crystal area to increase. The enhancement of the binding force between cellulose molecules improves its thermal stability and the temperature required for adsorption and hydrolysis. It was noteworthy that after 100 h of UV radiation, the endothermic peak (temperature) of dyed holocellulose increased, while the heat absorption (heat flow) decreased (Figure 4a).

In the process of UV radiation, the dyed holocellulose absorbs the energy of light, and part of it was converted into heat energy. The adsorbed water at different levels on the wood fiber cell wall was gradually dissociated from the outside to the inside. With the extension of radiation time, higher energy is required for the low-level adsorbed water desorption, resulting in the endothermic peak temperature of dyed holocellulose overall increases. The Siau model can explain the migration energy state (energy level) of water molecules adsorbed by multiple layers in the wood cell wall at different levels (Figure 4b) [18]. When the moisture content of wood is at the wood fiber saturation point, the enthalpy H of absorbed water is the same as that of liquid water (free water). However, when the wood moisture content is lower than the fiber saturation point, water molecules need to absorb more energy to migrate at different levels to overcome the energy difference between the two adsorption levels. Therefore, when the wood moisture content becomes lower, the desorption energy required for absorbed water in the wood becomes greater. However, the short wavelength UV light has destroyed part of the wood fiber’s internal structure, making the crystal area of the fiber decrease. This induces the thermal stability of the wood fiber decrease, resulting in a reduction in the heat flow required for the absorbed water desorption.

After dyed wood fiber absorbs the energy of UV light, partial energy may be converted into heat, and heat can improve the movement activity and oxidation reaction rate of wood fiber molecules. When the molecule’s kinetic energy exceeds the chemical bond dissociation energy, it may cause the thermal degradation or chemical group shedding of the cellulose molecular chain. This resulted in changing the aggregation structure of the cellulose macromolecular chain, reducing the wood’s relative crystallinity, and making the fiber material soft and weak [16,17]. It can be concluded that the thermal action could further synergize the photooxidation and degradation of dyed wood fibers.

### 3.4. Effect of UV Radiation on the Chemical Structure of Dyed Holocellulose

Cellulose is the main component of the wood cell wall, and hydroxyl is the most prominent infrared sensitive group of cellulose. The secondary alcohol hydroxyl at the C_2_ and C_3_ positions and the primary alcohol hydroxyl at the C_6_ positions in the cellulose sugar unit are free hydroxyl groups, which have high reactivity and can be subject to oxidation, esterification, etherification, and other reactions. The macromolecular chains of cellulose in the crystalline region are arranged regularly, and the reactivity of hydroxyl groups is lower than that in the amorphous region. The lower the crystallinity of wood fiber, the more active hydroxyl groups, and the higher the reactivity of dyed wood; hemicellulose is also a linear natural polysaccharide, containing acetyl, carboxyl, and other infrared sensitive groups.

The infrared spectrum of dyed holocellulose after UV radiation have not produced new absorption bands (Figure 5), and the intensity of each characteristic peak changed little, indicating that no new chemical functional groups were created on the molecular chain of dyed holocellulose during the light radiation. The chemical functional groups characterized at 1370 cm^−1^, 1160 cm^−1^, and 895 cm^−1^ were almost unaffected by UV radiation, indicating that holocellulose was relatively stable to UV light, and it has little contribution to the light degradation of dyed wood in the short term.

However, comparing the spectra before and after UV radiation, it can be found that the characteristic peak intensity of O-H hydroxyl stretching vibration at 3340 cm^−1^ and C-O alkoxy bond stretching vibration at 1050 cm^−1^ were weakened after UV radiation, while the characteristic peak intensity of C=O carbonyl stretching vibration at 1730 cm^−1^ has enhanced. This shows that cellulose undergoes oxidation reactions during light radiation. The free hydroxy-OH at C_2_, C_3_, C_6_ or C_2_, C_3_ reacted respectively or simultaneously to produce carbonyl C=O, indicating that hydroxyl radicals destroyed the intramolecular and intermolecular hydrogen bonds of cellulose, but also oxidized the glucosyl ring in the cellulose molecular chain *β*-alkoxycarbonyl structure. However, this structure was usually unstable and easy to pass *β*- alkoxy elimination reaction, resulting in the C-O of the *β*-1,4 glycoside bond being oxidized and broken, thus reducing the degree of polymerization (depolymerization) of the cellulose macromolecular chain. With the extension of UV radiation, the decomposition products (aldehydes, ketones, etc.) produced by the *β*-alkoxy elimination reaction can be further oxidized and decomposed into a series of organic acids (such as arabinoic acid, pyruvic acid, glucuronic acid, formic acid, glyoxylic acid, etc.) [18,19,20], which ultimately leads to the peeling, degradation and aging of cellulose.

In addition, after UV radiation, the intensity of the C-O stretching vibration characteristic peak for hemicellulose at 1050 cm^−1^ and the characteristic peak of bending vibration in the O-H plane at 1205 cm^−1^ also decreased slightly. This indicates that hemicellulose in dyed wood has also undergone oxidation degradation to a certain extent under UV radiation.

After UV radiation, the C=O bond characterized at 1442 cm^−1^ and 1597 cm^−1^ increased significantly, indicating that the acid dyes were prone to photooxidation and decomposed into colorless ketone structure (Figure 6). The change and destruction of the color system lead to the discoloration of the dyed wood surface. 

In addition, in the early stage of light radiation (Table 1, the first 20 h), although the acid dyes particles adsorbed on the wood surface first began to deteriorate, they also protected the wood structure to a certainty. However, with the extension of radiation time, the dyes particles degenerated into relatively stable small molecules, and then the wood fiber tissue also began to deteriorate under UV irradiation. It resulted in decreasing of the wood fiber polymerization degree and relative crystallinity. Therefore, the structure and stability of dyes affect the photochromism of dyed wood. Choosing dyes with good light fastness is beneficial to protect the dyed wood fiber structure, thus extending its service life.

It can be concluded that long-term UV radiation can lead to the slow oxidative degradation of the dyed wood holocellulose, oxidized cellulose, and strong reducing organic matter was formatted. Some fiber macromolecular chains were broken, further leading to the fiber microfilaments peeling, and the wood fiber discoloration after oxidation. In addition, other components in the color system of dyed wood, such as lignin, extracts, free radicals, and acid products produced by the light degradation of dyes, may also promote the photochemical degradation of wood holocellulose.

### 3.5. Effect of UV Radiation on the Microscopic Morphology of Dyed Wood

The surface micro-morphology of dyed wood fibers before and after UV radiation has not changed significantly (Figure 7), and there was no serious light degradation, indicating that the dyed wood fibers were relatively stable to light. However, after 100 h of UV radiation, some fibers shown obvious photolithographic characteristics, and there were pits or peelings in some parts, which indicated that dyed wood fibers underwent particular damage under UV radiation. This was mainly because photochemical or photooxidation reactions change the chemical composition of the dyed wood polymer system. In the beginning, the chemical bonds on the fiber surface break, and then dense pits appear on the fiber surface. The fiber microfilaments or bundles show peeling or partial peeling. With the UV radiation time extension, the phenomenon of light degradation gradually penetrated into the fiber interior through the free radical reaction transfer. This caused severe defects in the interior of the dyed wood fiber, resulting in fiber splitting, partial fracture and/or shedding, and morphology destruction [21,22]. Therefore, long-term light radiation lead to the deterioration and corrosion of dyed wood fibers, and the surface of dyed wood becomes rough and cracked, thus affecting its processing and finishing performance.

## 4. Conclusions

Dyeing treatment can effectively improve the surface decoration performance and commercial value of plantation fast-growing wood. It is beneficial to reduce timber cutting and protect the forest’s ecological environment. However, the dyed wood is prone to photochemical fading, restricting its popularization and application. Holocellulose is the main chemical component of dyed wood. Studying the photodegradation behavior of dyed holocellulose under UV radiation is conducive to clarifying the photochromic mechanism of dyed wood, promoting the efficient use of fast-growing wood, and the development of the wood dyeing industry. In this paper, the changes in the chemical structure and micromorphology of dyed wood and holocellulose under UV radiation were discussed. It is helpful in understanding the photochromic mechanism of dyed wood. It can also provide theoretical guidance for preventing the light degradation of dyed wood for interior decoration and improving its weather resistance. The main conclusions are as follows.

(1) UV radiation has little effect on the lattice structure of dyed wood fibers. The diffraction angle 2*θ* and the distance between layers of wood fiber crystal zone have remained basically unchanged. The dyed wood fiber was relatively stable to UV radiation.

(2) With the UV radiation time extension, the relative crystallinity and average crystallinity of dyed wood and holocellulose increased first and then decreased, but the overall change was insignificant. The change range of the relative crystallinity of dyed wood was no more than 3%, and that of holocellulose was not more than 5%.

(3) UV radiation caused the molecular chain chemical bond in the non-crystalline region of dyed cellulose to break, the fiber underwent photooxidation degradation, and the surface photoetching feature was prominent. The damage and destruction of wood fiber morphology led to the degradation of dyed wood.

## Figures and Tables

**Figure 1 polymers-15-01125-f001:**
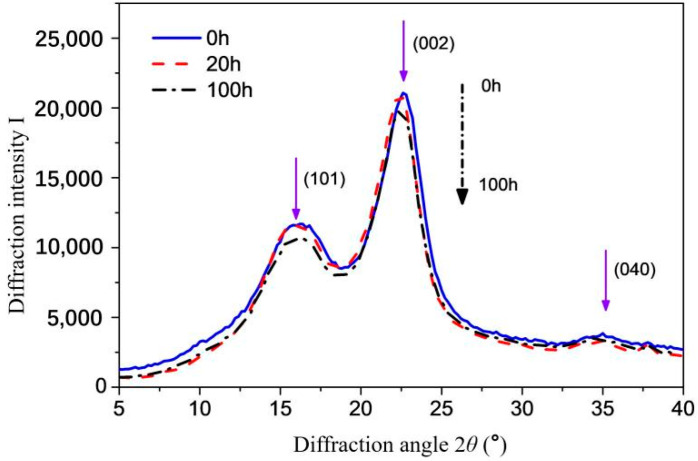
XRD curves of dyed wood powder during UV irradiation.

**Figure 2 polymers-15-01125-f002:**
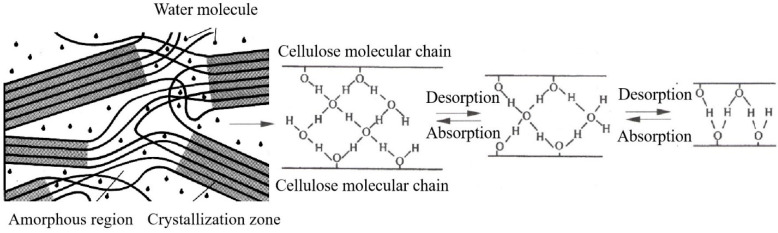
Changes of bound water between microfibril of the non-crystalline area in wood cell walls.

**Figure 3 polymers-15-01125-f003:**
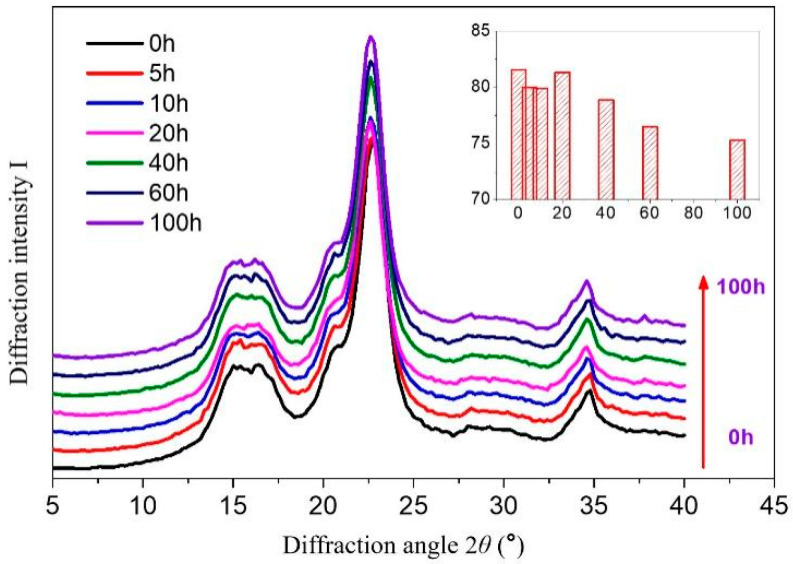
Relative crystallinity of dyed holocellulose during UV irradiation.

**Figure 4 polymers-15-01125-f004:**
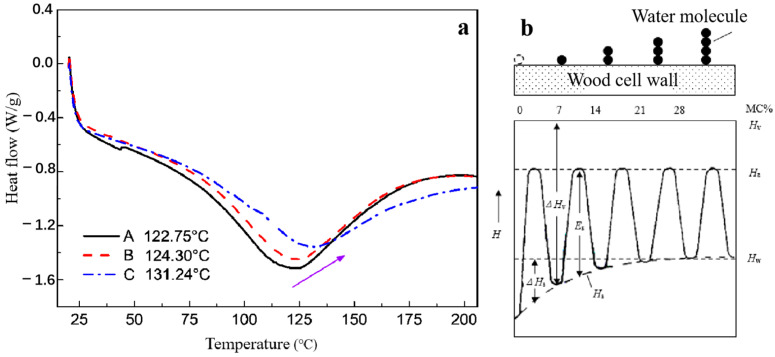
DSC curves of (A) holocellulose, (B) dyed holocellulose, and (C) dyed holocellulose after 100 h UV irradiation (**a**), and different level of water molecules adsorption model on wood cell wall (**b**).

**Figure 5 polymers-15-01125-f005:**
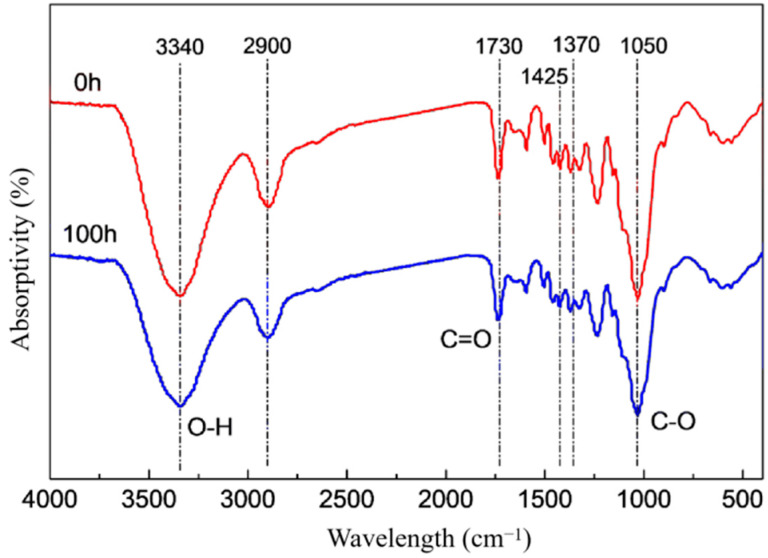
Chemical structure changes of dyed wood holocellulose before (the red line) and after (the blue line) UV radiation.

**Figure 6 polymers-15-01125-f006:**
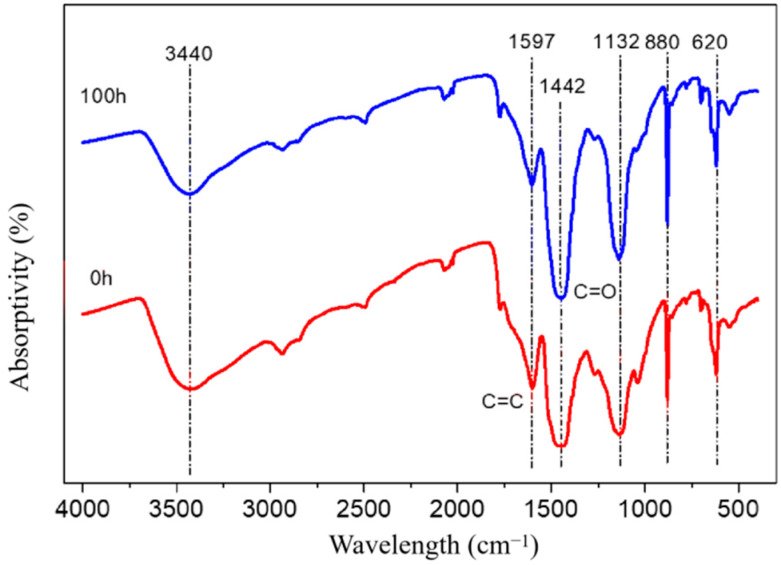
Infrared spectra of acid lake blue V dyes before (the red line) and after (the blue line) UV radiation.

**Figure 7 polymers-15-01125-f007:**
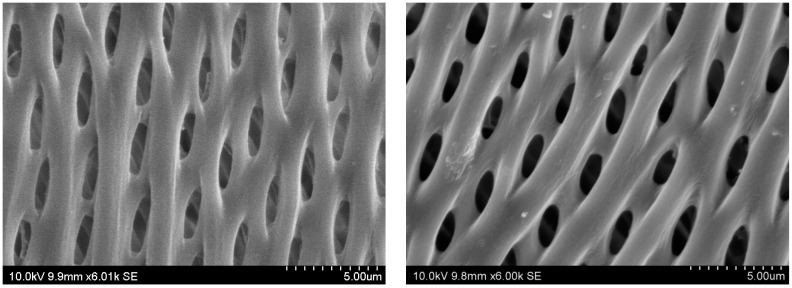
Surface micrographs of dyed wood before and after UV radiation.

**Table 1 polymers-15-01125-t001:** Crystallization parameters of dyed wood during UV irradiation.

Radiation Time (h)	*C_r_I* (%)	2*θ*_(002)_ (°)	*d*_(002)_ (nm)	FWHM_(002)_ (nm)	*D*_(002)_ (nm)
0	45.13	22.38	3.97	4.07	0.72
1	45.42	22.34	3.98	4.08	0.81
3	46.67	22.12	4.02	4.19	2.18
5	47.21	22.12	4.02	4.28	2.18
10	47.49	22.05	4.03	4.26	2.76
20	47.82	22.12	4.02	4.13	2.18
40	47.50	22.10	4.02	4.12	2.57
60	46.25	22.27	3.99	4.18	1.01
100	45.95	22.22	4.00	4.07	1.23

## Data Availability

The data presented in this study are available on request from the corresponding author.

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
