# Peer review of "Chemical Structure and Microscopic Morphology Changes of Dyed Wood Holocellulose Exposed to UV Irradiation"

_polymers, 2023, doi:10.3390/polym15051125_

Round 1

Reviewer 1 Report

In this paper, the authors have proposed the effect of UV irradiation on the chemical structure and microscopic morphology of wood holocellulose. However, the discussions for this version to support the big achievement of these fields are weak. Therefore, the authors need the revision the manuscript for publication in Polymers journal. We strongly believe that authors should enhance the research manuscript by improving readability and English skill and adding additional data and discussion for publication in Polymers journal. Some questions and suggestions are as followed;

[1] The Introduction section is too weak. We suggest that authors should add more background information and related literatures in the Introduction section. The Introduction section consists of paragraphs answering the following five questions: What is the problem? Why is it interesting and important? Why is it hard? Why hasn't it been solved before? (or, what's wrong with previously proposed solutions? What are the key components of my approach and results?

[2] We suggest that authors should change UV irradiation time to UV irradiation energy (intensity × time) for the realization and implementation of this study by journal readers.

[3] We suggest that authors should perform additional experiments on the revised manuscript, since the physicochemical properties of wood cellulose are critical factors before and after UV irradiation to increase understanding of this manuscript by journal readers.

[4] We suggest that authors should improve the quality of the Discussion section including deeper discussion in order to enhance the understanding of this manuscript by journal readers.

[5] What is the novelty in your work, please explain?

[6] Conclusions should be more concrete and future research directions presented. We suggest that authors are required to conclude the manuscript with their outputs in a quantitative manner. The value should be available to support this study.

[7] We judged that authors should cite the related literatures including recent literatures in the References section.

[8] The form of references described in the References part does not match the guideline of the “Polymers” journal. The authors should revise the references’ form accurately.

[9] We suggest that authors should improve English expression in the whole manuscript as much as possible.

Author Response

We thank the reviewers for their constructive evaluations and helpful comments on our work. We have modified the manuscript taking account into the reviewer’s suggestions. Point-by-point responses to all of the reviewers’ comments are given in the attachment file.

Reviewer 2 Report

Overall, this is a complete and meaningful work in dyed wood materials research. The effect of ultraviolet irradiation on chemical structure and microscopic morphology changes of dyed wood holocellulose were studied, it is helpful in understanding the photochromic mechanism of dyed wood. In general, the manuscript is well organized, both experiment design and results are correct. It is suggested to accept this paper subject to the following minor corrections.

1.In the part of abstract, the author should briefly explain the background at the beginning and summarize the research significance at the end to highlight the value of this study. 

2.Line 75, what is the molecular formula of acid dyes, and what are their characteristics or representativeness?

3.Line 105, what is the illuminance in the ultraviolet accelerated aging chamber?

4.What is the principle of acid dye dyeing, and what is the combination state of dye and wood?

5.In figure 6. The chemical functional groups or structural changes of acid dyes before and after ultraviolet radiation need to be more specific.

6.It is better to quote more relevant publications and references.

Author Response

(The authors gave the same response as above.)

Round 2

Reviewer 1 Report

The authors tried to improve the quality of this manuscript. Therefore, we recommend that this manuscript in the current version is publishable in polymers.